# Clinical and Biological Adaptations in Obese Older Adults Following 12-Weeks of High-Intensity Interval Training or Moderate-Intensity Continuous Training

**DOI:** 10.3390/healthcare10071346

**Published:** 2022-07-20

**Authors:** Layale Youssef, Jordan Granet, Vincent Marcangeli, Maude Dulac, Guy Hajj-Boutros, Olivier Reynaud, Fanny Buckinx, Pierrette Gaudreau, José A. Morais, Pascale Mauriège, Gilles Gouspillou, Philippe Noirez, Mylène Aubertin-Leheudre

**Affiliations:** 1École de Kinésiologie et des Sciences de l’Activité Physique, Université de Montréal, Montreal, QC H3T 1J4, Canada; layale.youssef@umontreal.ca; 2Centre de Recherche de l’Institut, Universitaire de Gériatrie de Montréal, Montreal, QC H3W 1W5, Canada; jgranet93@gmail.com (J.G.); fanny.buckinx@uliege.be (F.B.); gilles.gouspillou@gmail.com (G.G.); 3Département des Sciences Biologiques, Faculté des Sciences, Université du Québec à Montréal, Montreal, QC H2X 1Y4, Canada; vincentmarcangeli@gmail.com (V.M.); maudedulac@hotmail.com (M.D.); oreynaud26@gmail.com (O.R.); 4Département des Sciences de l’Activité Physique, Faculté des Sciences, Université du Québec à Montréal, Montreal, QC H2X 1Y4, Canada; 5Department of Medicine, Research Institute of the McGill University Health Centre, Montreal, QC H4A 3J1, Canada; guyelhajj@gmail.com (G.H.-B.); jose.morais@mcgill.ca (J.A.M.); 6Centre de Recherche du Centre Hospitalier de l’Université de Montréal, Montréal, QC H2X 3E4, Canada; pierrette.gaudreau@umontreal.ca; 7Centre de Recherche de l’Institut Universitaire de Cardiologie et Pneumologie de Québec, Université Laval, Quebec City, QC G1V 4G5, Canada; pascale.mauriege@kin.ulaval.ca; 8Département de Kinésiologie, Université Laval, Quebec City, QC G1V 0A6, Canada; 9Groupe de Recherche en Activité Physique Adaptée, Montreal, QC H2X 1Y4, Canada; 10PSMS, UFR STAPS, Université de Reims Champagne Ardenne, 51100 Reims, France; 11T3S, Inserm, Université Paris Cité, 75006 Paris, France; 12Institut de Recherche Médicale et d’Épidémiologie du Sport (IRMES), INSEP, Université Paris Cité, 75012 Paris, France

**Keywords:** high-intensity interval training (HIIT), moderate-intensity continuous training (MICT), aging, obesity

## Abstract

Sarcopenia and obesity are considered a double health burden. Therefore, the implementation of effective strategies is needed to improve the quality of life of older obese individuals. The aim of this study was to compare the impact of high-intensity interval training (HIIT) and moderate-intensity continuous training (MICT) on functional capacities, muscle function, body composition and blood biomarkers in obese older adults. Adipose tissue gene expression and markers of muscle mitochondrial content and quality control involved in exercise adaptations were also investigated. Sixty-eight participants performed either HIIT (*n* = 34) on an elliptical trainer or MICT (*n* = 34) on a treadmill, three times per week for 12 weeks. HIIT produced significantly higher benefits on some physical parameters (six-minute walking test (HIIT: +12.4% vs. MICT: +5.2%); step test (HIIT: +17.02% vs. MICT: +5.9%); ten-repetition chair test (HIIT: −17.04% vs. MICT: −4.7%)). Although both HIIT and MICT led to an improvement in lower limb power (HIIT: +25.2% vs. MICT: +20.4%), only MICT led to higher improvement in lower limb muscle strength (HIIT: +4.3% vs. MICT: +23.2%). HIIT was more beneficial for increasing total lean body mass (HIIT: +1.58% vs. MICT: −0.81%), while MICT was more effective for decreasing relative gynoid fat mass (HIIT: −1.09% vs. MICT: −4.20%). Regarding adipose tissue gene expression, a significant change was observed for cell death-inducing DFFA (DNA fragmentation factor-alpha)-like effector A (CIDEA) in the HIIT group (A.U; HIIT at T0: 32.10 ± 39.37 vs. HIIT at T12: 48.2 ± 59.2). Mitochondrial transcription factor A (TFAM) content, a marker of mitochondrial biogenesis, increased significantly following HIIT (+36.2%) and MICT (+57.2%). A significant increase was observed in the HIIT group for Translocase of Outer Membrane 20 (TOM20; +54.1%; marker of mitochondrial content), Mitofusin-2 (MFN2; +71.6%; marker of mitochondrial fusion) and Parkin RBR E3 Ubiquitin Protein Ligase (PARKIN; +42.3%; marker of mitophagy). Overall, our results indicate that even though MICT (walking on treadmill) and HIIT (on an elliptical) are effective intervention strategies in obese older adults, HIIT appears to have slightly more beneficial effects. More specifically, HIIT led to higher improvements than MICT on functional capacities, lean mass and skeletal muscle markers of mitochondrial content, fusion, and mitophagy. Thus, MICT but also HIIT (time-efficient training) could be recommended as exercise modalities for obese older adults to maintain or improve mobility, health and quality of life.

## 1. Introduction

Sarcopenia is the progressive loss of muscle function, mass and strength that naturally occurs with age [1]. The aging-related alterations in skeletal muscle biology increases the risk for older adults to become physically frail [2]. Another major public health issue is obesity, whose prevalence increases with aging. Sarcopenic obesity is considered a double health burden as it is associated with major adverse health outcomes including the development of frailty and disability, due to low muscle function and mass and an excess in adiposity levels [3]. There is therefore an urgent need to identify and implement effective strategies to improve the health status and quality of life of obese older adults.

Exercise training is recognized to provide major health benefits [4], and is considered an effective non-pharmacological strategy for older adults [5]. It has been observed that, after age 60, aerobic training as well as resistance training can be considered to maintain or improve muscle quality (strength/unit of muscle mass) [6]. Among exercise training modalities, aerobic moderate-intensity continuous training (MICT) is considered a successful intervention that reduces fat mass and metabolic disorders in obese adults [7]. In addition, high-intensity interval training (HIIT), a subtype of endurance training, is recognized as another time-efficient intervention for older adults, especially because lack of time is one of the main causes of inactivity in this population [8].

To make specific recommendations, studies have compared the clinical impact of HIIT and MICT in different populations. First, a systematic review comparing the effect of aerobic HIIT and MICT (average of three times per week for 12–16 weeks) in adults (age ≥ 18 years) with impaired vascular function reported that HIIT was more effective than MICT in improving brachial artery vascular function [9]. In obese individuals, a systematic review revealed that HIIT and MICT (interventions involving running and cycling for six to 16 weeks, with a frequency of two to five times per week) led to similar reductions in whole-body fat mass [10,11]. Additionally, a similar adherence and enjoyment were reported [12], as well as a similar improvement in insulin sensitivity [13]. When considering patients with coronary artery disease, a meta-analysis revealed that both exercise interventions (HIIT and MICT involving cycling and running) similarly improved glucose levels [14]. Finally, in non-obese older adults, it has been demonstrated that both training modalities (ergometer, four sessions per week for eight weeks) induced similar improvements in aerobic fitness (VO2 max) [15], although HIIT led to higher upper limb strength as well as superior body mass index (BMI) adaptations [16].

Thereafter, studies have compared molecular adaptations (muscle and adipose tissue (AT) metabolisms) following HIIT and MICT in different populations. For example, adaptations in AT metabolism are expected since aerobic exercise is already well known for its ability to reduce body fat. Indeed, it has been observed that endurance training is associated with an increase in catecholamine-induced lipolysis in obese humans [17] and AT lipoprotein lipase (LPL) activity in non-obese adults [18]. Finally, it has been observed that the circulating adipokine concentration as well as AT gene expression (uncoupling protein-1 (UCP1), cell death-inducing DFFA (DNA fragmentation factor-alpha)-like effector A (CIDEA) etc.) can be increased following exercise interventions [19]. In addition, the investigation of mitochondrial health following HIIT has recently been an area of interest. For example, exercise training appears to be an effective non-pharmaceutical means of maintaining muscle health by enhancing the skeletal mitochondrial content [20] and preserving mitochondrial health in older adults [21]. In young healthy adults, it was previously demonstrated that nuclear abundance of peroxisome proliferator-activated receptor-gamma coactivator (PGC1-α; a stimulator of mitochondrial biogenesis) in human skeletal muscles increased following an intense HIIT [22]. Moreover, mitochondrial content and oxidative phosphorylation (OXPHOS) capacity in skeletal muscles increased following HIIT (cycling three times per week for six weeks) in young overweight participants [23] and overweight older adults [24]. Additionally, 12 weeks of HIIT was also shown to increase multiple markers of mitochondrial content and quality control processes in obese older adults [25]. When comparing the effect of HIIT and MICT on the muscles’ mitochondrial content in overweight and obese young adults, it was reported that 10 weeks of cycling HIIT (twice per week) was superior to MICT for increasing mitochondrial content [26]. Furthermore, in young obese adults, mitochondrial respiration similarly improved after 12 weeks (three times per week) of HIIT and MICT on a treadmill [27].

Interestingly, to our knowledge, no study comparing HIIT to MICT explored the clinical, biological and molecular adaptations together (in the same study), nor in obese older adults. Therefore, the aim of this study was to compare the impact of HIIT and MICT on functional capacities, muscle function, body composition and blood biomarkers in obese older adults. A subset of participants underwent AT and muscle biopsies to investigate AT gene expression and markers of mitochondrial content and quality control involved in exercise adaptations.

## 2. Materials and Methods

### 2.1. Study Design

This study is an a posteriori study. The ethics committee of the “Université du Québec à Montréal (UQAM)” approved all procedures (#2014_e_1018_475). The participants provided their informed written consent after being informed about the study’s purpose, aim, procedures and associated risks. The participants included in the HIIT group were part of a previous study [25], and were matched by age and sex to the participants from another study which performed a MICT.

### 2.2. Participants

Participants were recruited from the community via social communication (flyers and meetings in community centers) in Greater Montreal. To be included in this study, participants had to meet the following criteria: (1) age 60 and over; (2) obese (BMI between 30 and 40 kg·m^−2^ or fat mass (%; DXA) equal or superior to 27% in men and 40% in women) or a waist circumference greater than 102 cm for men and 88 cm for women; (3) inactive (less than two hours of structured physical activity per week); (4) no involvement in a vigorous exercise program for at least 12 months; (5) able to follow the exercise training; (6) stable weight (±5 kg) for 6 months; (7) non-smokers and moderate drinkers (max: 15 g/day of alcohol); (8) able to understand French or English; and (9) postmenopausal for women (i.e., 12 consecutive months without menses). Exclusion criteria were the following: (1) presence of metal implant (pacemaker); (2) asthma requiring oral steroid treatment; (3) use of medication that could affect metabolism or cardiovascular function; (4) use of anticoagulants (only for participants undergoing biopsies). Participants with diagnosed but untreated neurological, cardiovascular or lung diseases, or cognitive disorders were also excluded.

Sixty-eight participants completed the intervention and were matched according to age (+/−2 years) and sex so that the two training modalities could be compared (HIIT: *n* = 34 vs. MICT: *n* = 34). Among these participants, 30 received subcutaneous abdominal AT biopsies (HIIT: *n* = 19 vs. MICT: *n* =11) and 25 received skeletal muscle biopsies (HIIT: *n* = 11 vs. MICT: *n* = 14) pre- and post-intervention. To be considered as having completed the intervention, participants had to complete at least 80% of the training sessions (minimum: 29/36 sessions) and perform the evaluation pre- and post-intervention.

### 2.3. Exercise Intervention

All the participants performed three supervised training sessions per week during 12 consecutive weeks.

#### 2.3.1. High-Intensity Interval Training (HIIT)

Participants performed their HIIT training on an elliptical device (TechnoGym Synchro Exc 700, Technogym, NJ, USA) to avoid impact and injuries and were supervised by trained kinesiologists (i.e., certified exercise instructors). The duration of the exercise session was 30 min and divided as follows: (1) five minute warm-up at a low intensity (50–60% maximal heart rate (MHR) and/or 8–12 on Borg’ scale); (2) twenty minutes of HIIT consisting of multiple 30-s high-intensity sprints (80–85% MHR or >17 on Borg’ scale) alternated with 90 s at moderate intensity (65% MHR or 13–16 on Borg’ scale); and (3) five-minute cool down period (50–60% MHR or 8–12 on Borg’ scale). MHR percentage and/or perceived exertion (Borg scale; relying exclusively on perceived exertion for participants using anti-arrhythmic and inotropic agents) were used to determine the intensity of each cycle. The following equation was used to determine the MHR: (((220-age)−Heart Rate rest) × % Heart Rate target) + Heart Rate rest. Speed and resistance of the elliptical device were continuously adjusted during the intervention to ensure that the MHR was always above 80% during high intensity intervals.

#### 2.3.2. Moderate-Intensity Continuous Training (MICT)

Participants followed a MICT, where they walked on a treadmill (Precor C936i, Precor, WA, USA) and were supervised by trained kinesiologists (i.e., certified exercise instructors). The MICT was performed at a moderate intensity (60–70% MHR or 13–14 Borg’s scale) for one hour per session. Speed and resistance of the treadmill were continuously adjusted during the intervention to ensure that the MHR was always between 60–70% MHR or 13–14 on Borg’s scale.

### 2.4. Physical Performance

Validated tests were used to evaluate physical performance and were previously described in Buckinx et al. [28]. The five tests used are briefly detailed below:

Six-minute walk test (6MWT): aerobic endurance was determined using the validated 6MWT [29,30] following the American Thoracic Society guidelines. Participants were instructed to walk at their own pace in an enclosed, flat 30-m-long track for six minutes. They were allowed to stop and rest as needed. The total distance covered at each minute and at the end was recorded.

Walking speed: This validated test [31,32] was conducted on an eight-meter straight line track. The two meters at the beginning and the end of the track, which included acceleration and deceleration phases, were not taken into account when calculating speed. The time taken (s) to complete the four meters at a usual and fast walking pace were recorded.

Unipodal balance: This test has extensive clinical support with a good test-retest reliability [33]. To assess static balance, participants were asked to stand on one leg with their eyes open and arms by their sides. The time was recorded in seconds from the moment one foot was lifted from the ground until it touched the ground again, the stance leg moved, or until 60 s had elapsed.

Timed up & go (TUG): This validated test [34] consists in standing from a sitting position on a chair, walking a three-meter distance and sitting down again. The test aims to estimate gait speed at a comfortable self-paced (TUG) speed as well as a fast-paced walking speed (TUGf). This test is recognized to predict fall risk.

Chair stand test (ten-repetitions): This test measures functional lower-body strength. This test is reproducible and correlates with lower extremity muscle strength [35]. Subjects were asked to stand up from a sitting position and to sit down ten times as fast as possible, with their arms folded across their chest. The time (in seconds) to complete this task was recorded.

The step test: This reliable and valid test [36,37] evaluates dynamic balance during an activity where the participant is required to be in movement and shift their body weight while standing on one leg. The participants were asked to perform the task as fast as possible for 20 s. The number of times the participant touched the top of the step with their foot was recorded.

### 2.5. Muscle Function Assessments

Loss of muscle strength and power are considered the main predictors of functional capacity decline as well as loss of mobility and autonomy. The three validated tests assessed to estimate muscle function were previously described in Buckinx et al. [28].

Grip strength: A hand dynamometer with an adjustable grip (Lafayette Instrument Company, Lafayette, IN, USA) was used to measure the maximum voluntary handgrip strength. To measure hand grip strength, participants were standing upright with the arm along the side of the body with the elbow extended and the palm of the hand facing the thigh. Participants were asked to squeeze the hand dynamometer as hard as possible for up to 4 s. This test was repeated three times for each hand, alternating between right and left, and the best result was recorded.

Lower limb muscle power: The Nottingham Leg Extensor Power Rig was used to assess lower limb muscle power while participants were seated. Participants were asked to use their dominant leg to push a pedal as fast and as hard as they could, which accelerated a flywheel.

Lower limb muscle strength: Participants were seated with the knee and hip joint angles set at 135° and 90° respectively. To measure strength, the tested leg was fixed to the lever arm at the level of the lateral malleoli on an analog strain gauge. The strongest of three maximum voluntary contractions was recorded.

These three muscle function measures were expressed in absolute (kg or W or N, respectively) and normalized to body weight and limb lean mass.

Anthropometric Characteristics

Body weight (kg) and height (m) were determined in fasting state using an electronic scale (GFK 660a, Adam Equipment Inc., Oxford, UK) and a stadiometer (Seca). Thereafter, BMI (body mass (kg)/height (m^2^)) was estimated.

### 2.6. Body Composition

Fat masses (total, android, gynoid, arm and leg, total; %) and total lean masses (total, arm and leg; kg) were quantified by dual-energy X-ray absorptiometry (DXA; GE Medical Systems, Madison, WI, USA) in fasted state.

### 2.7. Thigh Composition

A peripheral quantitative computed tomography (pQCT; Stratec XCT3000 system; STRATEC Medizintechnik GmbH), used at one third of the length of the right femur (distance from the lateral epicondyle to the greater trochanter), assessed thigh muscle composition (muscle area, subcutaneous and intramuscular fat contents; cm^2^). The total length of the femur, voxel size (0.5 mm) and speed (10 mm·s^−1^) were the scanning variables entered into the software. All scans were performed by operators trained in pQCT data acquisition according to Bone Diagnostics© guidelines (Fort Atkinson, WI, USA). Results were provided automatically in the ImageJ analysis (version 1.3.11; Bethesda, Rockville, MD, USA). For muscle and fat area, ranges for precision errors were reported to be between 2.1 and 3.7% and 2.4 and 6.4%, respectively [38].

### 2.8. Blood Parameters

After an overnight fast of 12 h, 15 mL of blood was collected from each participant to assess fasting serum levels of biochemical and hormonal markers. Participants were veni-punctured and blood was collected in gold vacutainer tubes (Becton-Dickinson, Frankli, NJ, USA). More specifically, the lipid profile (Total, HDL- and LDL-cholesterol, and TG levels), AT metabolites and adipokines (free fatty acids, adiponectin and leptin levels, adiponectin/leptin ratio), growth hormones (IGF1; IGFBP3 and IGFB3/IGF1 molar ratio) and glucose-insulin homeostasis (glucose and insulin levels but also HOMA and QUICKI indices) were assessed (see [25] for more details).

### 2.9. Adipose Tissue Biopsies and Quantification of Gene Expression

As previously described in Marcangeli et al. [25], biopsy samples were collected from an area in the lower quadrant (10–12 cm from the umbilicus) using a 12-gauge Yale needle. Abdominal subcutaneous AT samples (≈1 g) were obtained from the peri-umbilical region. The collected samples were immediately frozen in liquid nitrogen and were kept at −80 °C until further analysis of key gene expressions of the AT lipid metabolism. Genes selected represented important processes expected to be involved in reduced body weight and fat mass, as well as the conversion of white to brown AT in response to physical exercise.

### 2.10. Skeletal Muscle Biopsies and Immunoblotting

Skeletal muscle samples were obtained from the vastus lateralis muscle using Bergstrom needle biopsy. Muscle samples were frozen in liquid nitrogen and kept at −80 °C until further analysis. Immunoblots were performed to assess the content of multiple markers of mitochondrial biogenesis, content fusion, fission and mitophagy as extensively described in [25].

### 2.11. Energy Balance

Dietary intake: As previously described and validated in older adults, dietary intake was assessed before and after the intervention using the three-day food record method (two weekdays and one weekend day) [39]. Participants were asked to maintain their regular dietary habits during the intervention period. Analyses of total energy intake were performed using the Nutrific® web-applications (U Laval, Quebec city, QC, Canada).

Physical activity level: The number of steps was used to estimate the level of physical activity of participants using a validated tri-axial accelerometer SenseWear^®^ Mini Armband (BodyMedia, Pittsburgh, PA, USA) as previously described by Colbert et al. [39]. Participants had to wear the device on their left arm at all times during three consecutive days, except when showering or swimming. Each participant had to wear the device at least 85% of the time to be included in the study.

### 2.12. Sociodemographic and Cognitive Assessment

The validated Montreal Cognitive Assessment (MoCA) was used to assess cognitive status [40]. In cases in which a subject had ≤12 years of education, an extra point was added to the total score [40].

### 2.13. Statistical Analyses

Quantitative results were expressed as means ± SD. The homogeneity of variances was assessed using Levene’s test. The delta changes (%) were calculated as (post-pre)/pre × 100. The time effect (intervention), group effect and their interaction (time*group effect) on the clinical and biological parameters were tested using a linear mixed-models approach (nlme package) with two-factor repeated measures ANOVA. Simultaneous tests for general linear hypotheses (emmeans package) were used for post-hoc analyses with a Bonferroni correction. All statistical analyses were performed using the software R (4.2; foundation for statistical computing, Vienna, Austria), and results were considered statistically significant at *p*-value < 0.05.

## 3. Results

### 3.1. Adherence and Baseline Characteristics

At the end of our intervention, the adherence level of the participants in both groups was considered high. More specifically, among the 36 sessions, participants completed an average of 34.9 sessions for the HIIT group (97%) and 34.6 sessions for the MICT group (96%). The baseline characteristics were similar for both groups (Table 1).

### 3.2. The Impact of HIIT and MICT on Functional Capacities

The impacts of HIIT and MICT on functional capacities are detailed in Table 2, and the delta changes are detailed in Appendix A. A time effect was observed for all the functional capacity parameters evaluated. A time*group interaction was observed for the six-minute walking test (*p* = 0.004), the step test (*p* < 0.0001), and the ten-repetition chair test (*p* = 0.006; Figure 1). These results indicate that these two training modalities are effective for improving all functional capacities in obese older adults. However, we observed a significantly higher benefit for HIIT than MICT on some physical parameters (six-minute walking test (HIIT: +12.4% vs. MICT: +5.2%); step test (HIIT: +17.0% vs. MICT: +5.9%); ten-repetition chair test (HIIT: −17.0% vs. MICT: −4.7%)). Additionally, a clinical impact was observed for the HIIT group for the six-minute walking test, where the distance increased by more than 50 m, the minimal increase in distance required for a change to be considered clinically significant [41]. Walking speed significantly increased in both groups, but a clinical impact was only observed for the HIIT group where walking speed increased by more than 0.1 m/s, the minimal increase in walking speed required for a change to be considered clinically significant [42].

### 3.3. The Impact of HIIT and MICT on Skeletal Muscle Function

The impacts of HIIT and MICT on skeletal muscle function are detailed in Table 2, and the delta changes are detailed in Appendix A. Absolute and relative upper (handgrip strength) and lower (maximal quadriceps strength) limb muscle strength as well as lower limb power were assessed.

Absolute (kg) and relative (kg/body weight) upper limb muscle strength did not significantly change following both exercise modalities. However, a time*group interaction was observed for absolute and relative lower limb muscle strength. These parameters increased significantly more for the MICT group than HIIT group (Quadriceps strength (HIIT: +4.3% vs. MICT: +23.2%); quadriceps/body weight (HIIT: +4.1% vs. MICT: +23.6%); quadriceps/lean leg mass (HIIT: +2.3% vs. MICT: +25.3%); Figure 2). Finally, both groups significantly improved lower limb muscle power [(HIIT: +25.2% vs. MICT: +20.4%); Figure 2].

These results indicate that MICT seems more effective than HIIT for improving lower limb muscle strength in obese older adults.

### 3.4. The Impact of HIIT and MICT on Body Composition

The impacts of HIIT and MICT on body composition are detailed in Table 3, and the delta changes are detailed in Appendix A. Anthropometry, fat and lean mass, as well as thigh muscle quality were measured to assess the impact of HIIT and MICT on body composition. Participants improved only total (HIIT: +1.6% vs. MICT: −0.8%) and leg (HIIT: 2.1% vs. MICT: −0.8%) lean muscle mass, and this improvement was greater in the HIIT than MICT group (Figure 3). Participants improved only leg (HIIT: −1.6% vs. MICT: −3.7%) and gynoïd (HIIT: −1.1% vs. MICT: −4.2%) relative fat mass, and this improvement was greater in the MICT than HIIT group (Table 3 and Figure 4). No other change was observed. Considering that lean mass increased for the HIIT group and fat mass decreased for the MICT group, both training modalities are effective for improving body composition. Regarding muscle composition measured using pQCT, total muscle area, total fat area, and subcutaneous fat area, significantly decreased following MICT only (Table 3 and Figure 4).

### 3.5. The Impact of HIIT and MICT on Blood Parameters

The impacts of HIIT and MICT on blood parameters related to muscle metabolism or metabolic syndrome are detailed in Table 4, and the delta changes are detailed in Appendix A.

The only significant time effect observed was for triglycerides, which significantly decreased (−8.8%) in the HIIT group. These results indicate that HIIT is effective to improve triglyceride levels in obese older adults.

### 3.6. The Impact of HIIT and MICT on AT mRNA Gene Expression

Messenger ribonucleic acid (mRNA) levels of several factors were measured (Figure 5) to assess the impact of HIIT and MICT on AT gene expression. A significant increase in CIDEA gene expression was observed following HIIT only. Although there was already a difference regarding CIDEA gene expression at baseline (HIIT: 32.1 ± 39.4 vs. MICT: 16.1 ± 37.1 A.U., *p* = 0.01), a significantly higher increase for this gene was observed in the HIIT group compared to the MICT group at the end of the intervention (HIIT: 48.2 ± 59.2 vs. MICT: 14.4 ± 23.1 A.U., *p* < 0.001).

### 3.7. The Impact of HIIT and MICT on Skeletal Muscle Mitochondrial Content and Quality Control

The impacts of HIIT and MICT on skeletal muscle mitochondrial content are detailed in Table 5, and the delta changes are detailed in Appendix A. Several mitochondrial proteins were measured to assess the impact of HIIT and MICT on markers of mitochondrial content (translocase of outer membrane 20; TOM20), biogenesis (transcription factor A mitochondrial; TFAM), fusion (mitofusin-2; MFN2), fission (dynamin-related protein 1; DRP1), and mitophagy (Parkin RBR E3 ubiquitin protein ligase; PARKIN). A significant time effect was reported for the mitochondrial TFAM, for TOM20, and PARKIN. More specifically, TFAM increased significantly following HIIT (+36.2%) and MICT (+57.24%) (Figure 6). A significant increase was observed for TOM20 (+54.14%), Mitofusin-2 (MFN2; +71.6%), and PARKIN (+42.32%) in the HIIT group only (Figure 6). Neither HIIT nor MICT significantly altered DRP1 content. These results indicate that HIIT is effective for improving markers of muscle mitochondrial content, fusion and mitophagy, although both exercise interventions are effective for improving mitochondrial biogenesis.

## 4. Discussion

Due to the aging population worldwide and the progressive increase in the prevalence of obesity, finding effective strategies to reduce body fat and increase muscle function is of particular importance for improving the quality of life of afflicted individuals and to reduce healthcare costs. In this regard, the potential beneficial impacts of two aerobic training modalities (HIIT and MICT) performed over 12 weeks were investigated in obese older adults. The adherence level for our interventions was comparable between both groups (97% for HIIT and 96% for MICT). This finding is in line with a previous study that has reported that adherence and enjoyment level were comparable for HIIT and MICT in older adults [43]. In addition, our study showed that changes in several parameters were specific to the type of exercise training. Indeed, MICT was more beneficial for decreasing relative gynoid fat mass and increasing lower limb muscle strength while HIIT resulted in greater improvement in functional capacities and greater increase in total lean body mass.

Functional capacities reflect the actions used in daily life and decreases with aging [44]. Interestingly, we report here that while both HIIT and MICT are effective in improving several functional capacities, HIIT led to greater improvement in the performance at several functional capacity tests (six-minute walking test, step test and ten-repetition chair test) in obese older adults. These results are in line with a previous study conducted in older women showing that functional capacities improved following both HIIT and MICT [45]. Additionally, our results showed that the lower limb power increased for both groups. The total and legs lean mass increased following HIIT, and the gynoid fat mass decreased following MICT. These different findings could be explained by the use of the elliptical trainer which produces more whole-body movement compared to the treadmill. In addition, HIIT and MICT led to greater improvement in lower limb power whereas MICT led to higher improvement in lower limb muscle strength. A possible explanation could be that a higher speed was used for HIIT (elliptical movements), and a higher force was used for MICT (impact when walking). Another possible explanation could be the presence of lower body weight strain during elliptical training compared to higher body weight strain on the treadmill. Contrarily to our findings, a study conducted in older adults showed that HIIT led to greater improvement in upper limb strength than MICT [16]. One of the reasons might be due to the higher handgrip strength values obtained for both groups at baseline in our intervention ((HS (kg): HIIT: 33.2 & MICT: 32.1)) compared to Ballesta-García et al. ((HS (kg): HIIT: 25.4 & MICT: 22.9)), even if the dynamometer used to assess grip strength and the population were different [17].

Regarding blood parameters, our results are supported by a meta-analysis revealing that triglyceride levels generally decrease after HIIT [46]. Additionally, in patients with coronary artery disease, glucose levels similarly improved following both HIIT and MICT [14], although no significant difference was observed following our intervention. A possible reason could be the different age range and obesity status of our participants that could differentially affect biological parameters. 

Concerning body composition, interesting differences between HIIT and MICT modalities were observed for fat mass and lean mass adaptations after our intervention. HIIT was more beneficial for increasing total lean body mass (HIIT: +1.58% vs. MICT: −0.81%), while MICT was more beneficial for decreasing relative gynoid fat mass (HIIT: −1.09% vs. −4.20%). These results are consistent with a previous study showing that total and visceral fat mass decreased after 12 weeks of MICT in obese older adults [47]. In young adults, it was previously found that HIIT had a greater effect on whole-body adiposity, while lower intensity training had a greater effect on subcutaneous abdominal and visceral fat mass [48]. Interestingly, in young obese adults, no significant difference in body adiposity was observed [10]. This discrepancy with our findings could be due to the age difference, where older obese adults could have different mechanisms regulating fat metabolism than younger obese adults. In line with our findings, a greater increase in lean mass after HIIT compared to MICT was observed in older patients undergoing cardiac rehabilitation [49]. Although HIIT and MICT differentially affected body composition, our results indicate that both training modalities can be considered effective in improving body composition in obese older adults. 

Regarding adipose tissue gene expression, a significant change was observed only for CIDEA, which is associated with lipid droplets and insulin sensitivity in humans and is considered an important regulator of fat metabolism [50]. Among all the parameters of gene expression, the reason why only CIDEA significantly changed following our intervention might be due to the insufficient changes in fat mass to induce molecular adaptations, since the changes were not clinically significant (delta change: <5%) [51]. Taken altogether, these data indicate that 12 weeks of HIIT or MICT had a limited impact on AT gene expression within the set of genes studied. These results suggest that the positive impact of MICT on gynoid fat mass decrease might not be related to transcriptional reprogramming in adipocytes, based on the limited set of genes evaluated. Multi-omics approaches will be required to confirm and extend our results on metabolic adaptations in human adipocytes following aerobic exercise interventions.

Accumulation of mitochondrial dysfunction is believed to play a key role in the muscle aging process [52]. Furthermore, intramuscular lipid accumulation in obese individuals has been linked to altered skeletal muscle mitochondrial content and function [53]. As we previously reported [25], HIIT effectively increased markers of mitochondrial biogenesis (TFAM), mitochondrial fusion (MFN2), mitochondrial content (TOM 20), and mitophagy (Parkin) in obese older adults. Additionally, an increase in TFAM was also observed for MICT. Interestingly, neither HIIT not MICT altered Drp1 content, a marker of mitochondrial fission. Previous studies performed in obese [26] and older [24] adults demonstrated that mitochondrial content improved more after HIIT compared to MICT. Interestingly, our study, which focused on older and obese individuals, reports similar findings, although differences between HIIT and MICT did not reach statistical significance.

To our knowledge, our study is the first to compare the impact of HIIT and MICT in obese older adults using a deep phenotyping approach. This approach involves an extensive clinical assessment using gold standard methods (functional capacities, muscle function, body composition, blood biomarkers) as well as an exploration of potential mechanistic explanations (AT gene expression and markers of mitochondrial content and quality control). However, our study has some limitations. First, we did not evaluate sex-specific adaptations induced by HIIT and/or MICT interventions since the sample size for molecular and cellular assessments were limited to ensure adequate statistical power. We performed a matched group per-protocol analysis to compare our interventions. The associated risk was therefore to overestimate the effects of our interventions. To confirm our encouraging results, intention-to-treat analysis should also be performed in future studies. Regarding participants, extrapolation of the results requires caution since our participants were moderately obese on average and hence may not directly translate to population that are either older or more severely obese. A selection bias is also possible as only volunteer subjects were included in the exercise interventions. Moreover, the two exercise modalities were not performed on the same device (elliptical trainer for HIIT and treadmill for MICT). The reason for this was to prevent joint injuries on the lower limbs, due to the high surface impacts that might be caused by HIIT, as some of our participants may have been suffering from osteoporosis. However, for future studies, it would be interesting to perform a RCT to confirm our promising results by adding one group performing MICT on the elliptical trainer and also an inactive control group. It will also be of great interest to compare our results with that of a younger population to assess if there is an age effect and to investigate in older people if greater adaptations are observed with a longer intervention. Additionally, following the numerous analyses done in this study, it would be interesting to conduct serum metabolomic analyses before and after both exercise modalities and evaluate the metabolomic signatures behind the physiological changes. Thus, new putative biomarkers specific to HIIT and MICT in obese older adults could be used for therapeutic strategies to treat obesity and age-related decline in muscle mass and function.

## 5. Conclusions

Overall, our results showed that even though MICT (walking on treadmill) and HIIT (on an elliptical) are effective intervention strategies to improve the health status of obese older adults, HIIT appeared to have slightly more beneficial effects. More specifically, MICT was more beneficial for decreasing relative gynoid fat mass and increasing lower limb muscle strength whereas HIIT led to better improvements than MICT on functional capacities, lean mass and skeletal muscle markers of mitochondrial biogenesis, content, fusion, and mitophagy. Thus, MICT and HIIT (which is more time-efficient since it requires half the time of MICT), could be recommended for obese older adults in order to maintain or improve mobility, health and quality of life.

## Figures and Tables

**Figure 1 healthcare-10-01346-f001:**
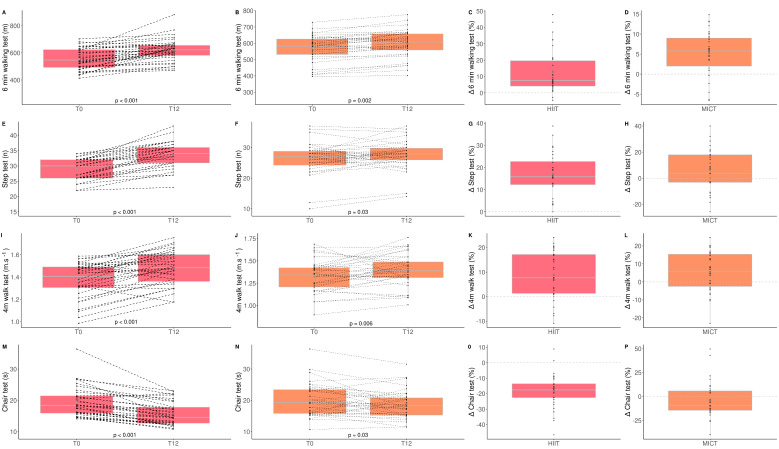
Impact of HIIT and MICT on functional capacities. HIIT = high-intensity interval training (pink), MICT = moderate-intensity continuous training (orange), T0 = before the 12-week intervention, T12 = after the 12-week intervention, delta change between T0 and T12 is expressed in %. six-min walking test for HIIT (**A**), six-minute walking test for MICT (**B**), delta change of six-minute walking test for HIIT (**C**), delta change of six-minute walking test for MICT (**D**), step test for HIIT (**E**), step test for MICT (**F**), delta change of step test for HIIT (**G**), delta change of step test for MICT (**H**), four-meter walk test for HIIT (**I**), four-meter walk test for MICT (**J**), delta change of four-meter walk test for HIIT (**K**), delta change for four-meter walk test for MICT (**L**), chair test for HIIT (**M**), chair test for MICT (**N**), delta change of chair test for HIIT (**O**), delta change of chair test for MICT (**P**). Each black dot represents a subject (the dashed line connects the T0 and T12 values of each subject), the grey point the mean, and the grey line in the boxplot the median.

**Figure 2 healthcare-10-01346-f002:**
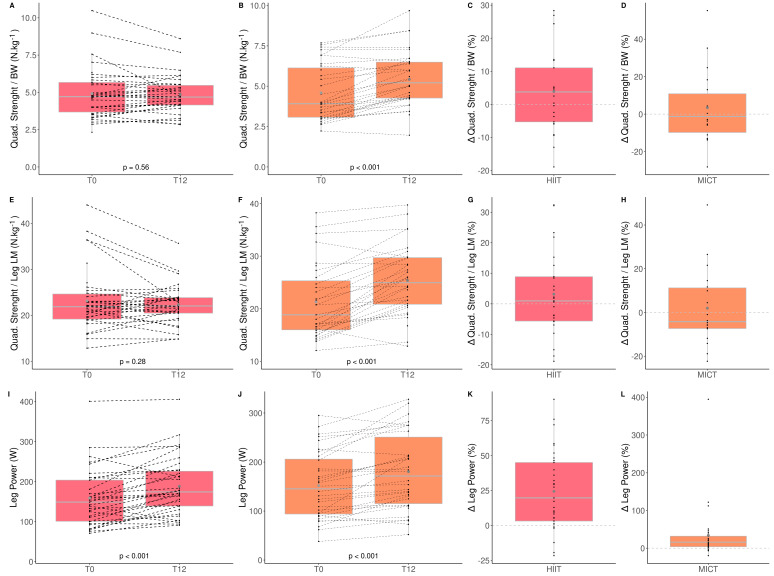
Impact of HIIT and MICT on skeletal muscle function. HIIT = high-intensity interval training (pink), MICT = moderate-intensity continuous training (orange), T0 = before the 12-week intervention, T12 = after the 12-week intervention, delta change between T0 and T12 is expressed in %. Quadriceps strength normalized to body weight for HIIT (**A**), Quadriceps strength normalized to body weight for MICT (**B**), delta change of quadriceps strength normalized to body weight for HIIT (**C**), delta change of quadriceps strength normalized to body weight for MICT (**D**), quadriceps strength normalized to leg lean mass for HIIT (**E**), quadriceps strength normalized to leg lean mass for MICT (**F**), delta change of quadriceps strength normalized to leg lean mass for HIIT (**G**), quadriceps strength normalized to leg lean mass for MICT (**H**), leg power for HIIT (**I**), leg power for MICT (**J**), delta change of leg power for HIIT (**K**), delta change of leg power for MICT (**L**). Each black dot represents a subject (the dashed line connects the T0 and T12 values of each subject), the grey point the mean, and the grey line in the boxplot the median.

**Figure 3 healthcare-10-01346-f003:**
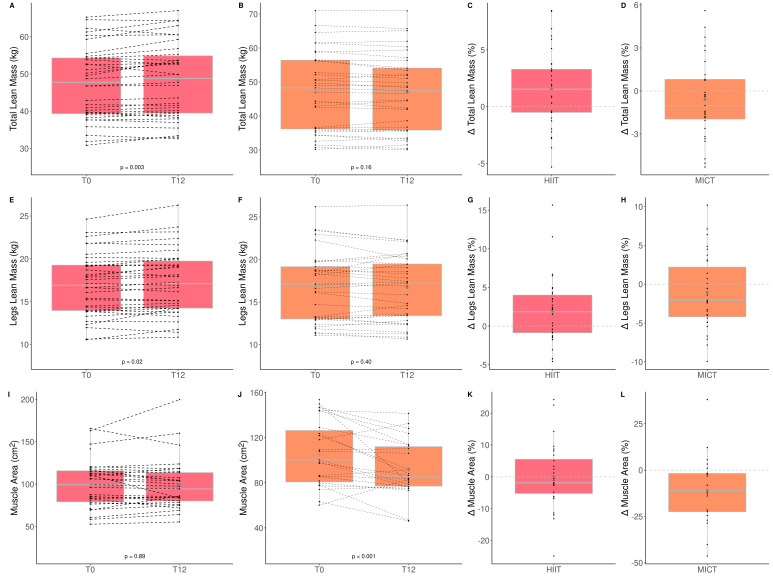
Impact of HIIT and MICT on lean mass. HIIT = high-intensity interval training (pink), MICT = moderate-intensity continuous training (orange), T0 = before the 12-week intervention, T12 = after the 12-week intervention, delta change between T0 and T12 is expressed in %. Total lean mass for HIIT (**A**), total lean mass for MICT (**B**), delta change of total lean mass for HIIT (**C**), delta change of total lean mass for MICT (**D**), legs lean mass for HIIT (**E**), legs lean mass for MICT (**F**), delta change of legs lean mass for HIIT (**G**), delta change of total legs lean mass for MICT (**H**), muscle area for HIIT (**I**), muscle area for MICT (**J**), delta change of muscle area for HIIT (**K**), delta change of muscle for MICT (**L**). Each black dot represents a subject (the dashed line connects the T0 and T12 values of each subject), the grey point the mean, and the grey line in the boxplot the median.

**Figure 4 healthcare-10-01346-f004:**
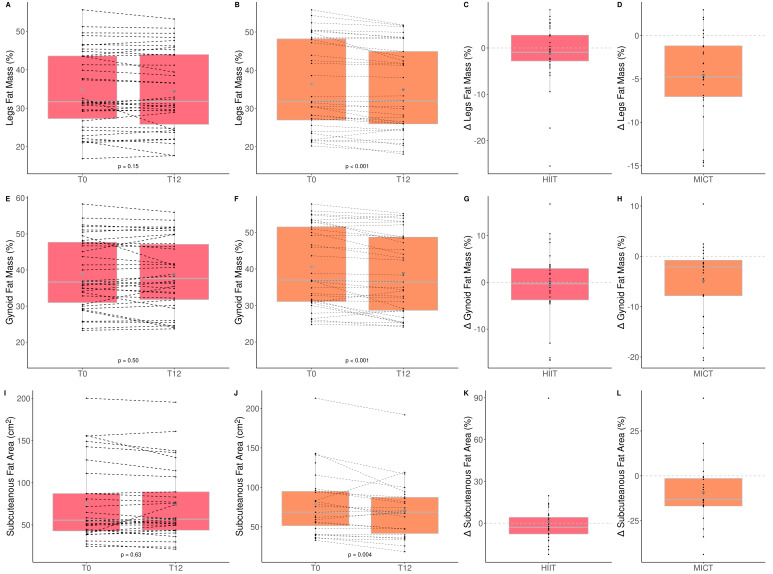
Impact of HIIT and MICT on fat mass. HIIT = high-intensity interval training (pink), MICT = moderate-intensity continuous training (orange), T0 = before the 12-week intervention, T12 = after the 12-week intervention, delta change between T0 and T12 is expressed in %. Relative legs fat mass for HIIT (**A**), relative legs fat mass for MICT (**B**), delta change of relative legs fat mass for HIIT (**C**), delta change of relative legs fat mass for MICT (**D**), relative gynoid fat mass for HIIT (**E**), relative gynoid fat mass for MICT (**F**), delta change of relative gynoid fat mass for HIIT (**G**), delta change of relative gynoid fat mass for HIIT (**H**), subcutaneous fat area for HIIT (**I**), subcutaneous fat area for MICT (**J**), delta change of subcutaneous fat area for HIIT (**K**), delta change of subcutaneous fat area for MICT (**L**). Each black dot represents a subject (the dashed line connects the T0 and T12 values of each subject), the grey point the mean, and the grey line in the boxplot the median.

**Figure 5 healthcare-10-01346-f005:**
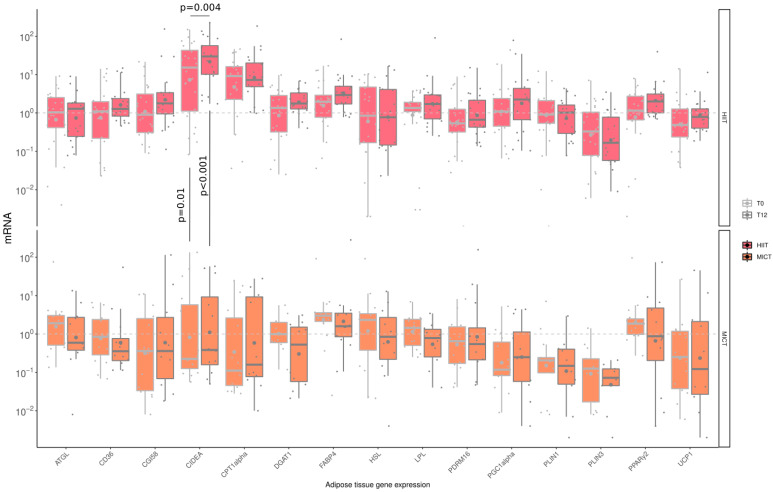
Impact of HIIT and MICT on adipose tissue mRNA gene expression. HIIT = High-Intensity Interval Training (pink), MICT = Moderate-Intensity Continuous Training (orange), T0 = before the 12-week intervention (light grey), T12 = after the 12-week intervention (dark grey), each dot represents a subject, the grey point the mean, and the grey line in the boxplot the median, ATGL = adipose triglyceride lipase; CD36 = cluster of differentiation 36; CGI58 = comparative gene identification-58; CIDEA = cell death-inducing DFFA-like effector a; CPT1-α Carnityl PalmitolylTransferase-1 α; DGAT-1 = diacylglycerol acyl transferase-1; FABP-4 = fatty acid binding protein-4; HSL = hormone sensitive lipase; LPL = lipoprotein lipase; PDRM16 = PR domain containing 16; PGC-1α = peroxisome proliferator-activated receptor-gamma coactivator-1α; PLIN1 = perilipin-1; PLIN3 = perilipin-3; PPAR γ2 = peroxisome proliferator-activated receptor γ2; UCP-1= uncoupling protein-1.

**Figure 6 healthcare-10-01346-f006:**
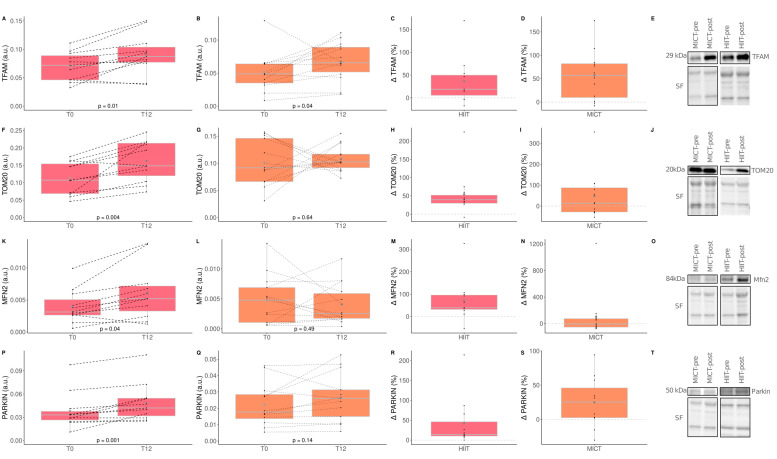
Impact of HIIT and MICT on markers of skeletal muscle mitochondrial content and quality control processes. HIIT = high-intensity interval training (pink), MICT = moderate-intensity continuous training (orange), T0 = before the 12-week intervention, T12 = after the 12-week intervention, delta change between T0 and T12 is expressed in %. TFAM = mitochondrial transcription factor A; TOM20 = translocase of outer membrane 20; MFN2 = Mitofusin-2; PARKIN = Parkin RBR E3 ubiquitin protein ligase. TFAM content in HIIT (**A**), TFAM content in MICT (**B**), delta change of TFAM content in HIIT (**C**), delta change of TFAM content in MICT (**D**), representative TFAM western blot and its corresponding stain free (loading control) for HIIT and MICT pre and post intervention (**E**), TOM20 content in HIIT (**F**), TOM20 content in MICT (**G**), delta change of TOM20 content in HIIT (**H**), delta change of TOM20 content in MICT (**I**), representative TOM20 western blot and its corresponding stain free (loading control) for HIIT and MICT pre and post intervention (**J**), MFN2 content in HIIT (**K**), MFN2 content in MICT (**L**), delta change of MFN2 content in HIIT (**M**), delta change of MFN2 content in MICT (**N**), representative MFN2 western blot and its corresponding stain free (loading control) for HIIT and MICT pre and post intervention (**O**), PARKIN content in HIIT (**P**), PARKIN content in MICT (**Q**), delta change of PARKIN content in HIIT (**R**), delta change of PARKIN content in MICT (**S**), representative PARKIN western blot and its corresponding stain free (loading control) for HIIT and MICT pre and post intervention (**T**). Each black dot represents a subject (the dashed line connects the T0 and T12 values of each subject), the grey point the mean, and the grey line in the boxplot the median.

**Table 1 healthcare-10-01346-t001:** Characteristics of participants at baseline.

	HIIT	MICT	*p*-Value
Age (years)	67.7 ± 4.6	68.1 ± 4.1	0.67
Men/women (n)	18/16	18/16	N/A
Fat mass (%—Men/Women)	32.0/42.4	32.2/43.6	0.90/0.51
BMI (kg/m^2^)	28.9 ± 29.0	29.8 ± 29.6	0.52
Total fat mass (%)	36.9 ± 7.6	37.4 ± 7.5	0.78
Android fat mass (%)	46.5 ± 7.52	46.5 ± 6.4	0.98
Gynoid fat mass (%)	39.0 ± 9.8	40.6 ± 10.9	0.53
Steps per day (n)	6476 ± 3180	6865 ± 3819	0.75
Energy intake (kcal/day)	2129 ± 796	2057 ± 464	0.67
MoCA (/30)	27.8 ± 1.5	27.3 ± 2.2	0.28

Data are presented as: mean ± SD. HIIT = High-Intensity Interval Training; MICT = Moderate-Intensity Continuous Training; BMI = body mass index; MoCA = Montreal Cognitive Assessment.

**Table 2 healthcare-10-01346-t002:** Functional capacities and skeletal muscle function in obese older adults following 12 weeks of High-Intensity Interval Training (HIIT) and Moderate-Intensity Continuous Training (MICT).

Parameters	HIIT (*n* = 34)		MICT (*n* = 34)		*p*-Value		
	Pre	Post	Pre	Post	Time Effect	Group Effect	Time*Group Effect
**Functional capacities**							
6 min walking test (m)	554 ± 79	620 ± 84 ***	570 ± 84	598 ± 85 **	**<0.0001**	0.88	**0.004**
Step test (*n*)	28.97 ± 3.53	33.60 ± 4.20 ***	26.35 ± 5.40	27.53 ± 4.81 *	**<0.0001**	**<0.0001**	**<0.0001**
4 m walk test normal (m/s)	1.37 ± 0.16	1.48 ± 0.15 ***	1.33 ± 0.18	1.39 ± 0.18 **	**<0.0001**	0.11	0.18
4 m walk test fast (m/s)	1.93 ± 0.21	2.10 ± 0.25 **	1.87 ± 0.30	2.02 ± 0.46 **	**0.0002**	0.26	0.82
Unipodal Balance test (s/60sec)	27.60 ± 17.67	38.51 ± 19.67	34.38 ± 20.60	38.70 ± 22.88	**0.0007**	0.44	0.12
Chair test (s)	19.57 ± 4.92	15.78 ± 3.72	20.11 ± 5.37	18.78 ± 4.50	**<0.0001**	0.09	**0.006**
Timed Up and Go Test (s)	10.07 ± 1.59	9.12 ± 1.22	10.07 ± 1.71	9.31 ± 1.97	**<0.0001**	0.79	0.60
**Skeletal muscle function**							
Handgrip strength (kg)	33.23 ± 11.77	34.32 ± 10.42	32.16 ± 10.60	32.80 ± 9.35	0.46	0.57	0.83
Handgrip strength/BW	0.41 ± 0.11	0.43 ± 0.09	0.40 ± 0.12	0.42 ± 0.10	0.32	0.70	0.89
Handgrip strength/ ALM	6.08 ± 1.94	6.27 ± 0.97	6.19 ± 1.04	6.73 ± 157	0.12	0.35	0.45
Quadriceps strength (N)	397 ± 174	388 ± 144	360 ± 160	425 ± 111 ***	**0.0001**	0.94	**<0.0001**
Quad/BW	4.92 ± 1.73	4.85 ± 1.30	4.58 ± 1.73	5.42 ± 1.70 ***	**<0.0001**	0.34	**<0.0001**
Quad/LLM	23.15 ± 6.93	22.34 ± 4.41	21.30 ± 7.08	25.36 ± 6.53 ***	**0.001**	0.78	**<0.0001**
Lower limb power (W)	159 ± 72	188 ± 72 ***	152 ± 69	180 ± 78 ***	**<0.0001**	0.67	0.86

Data are presented as: mean ± SD. HIIT = high-intensity interval training; MICT = moderate-intensity continuous training; Pre = before the 12-week intervention; Post = after the 12-week intervention; BW = body weight; ALM = arms lean mass; LLM = legs lean lass; Quad = quadriceps. Time effect, Group effect and Time*Group effect were analyzed using two-way repeated measures ANOVA. * *p* < 0.05, ** *p* < 0.01, *** *p* <0.001 = HIIT effect, and MICT effect (analyzed using post-hoc tests).

**Table 3 healthcare-10-01346-t003:** Body composition parameters in obese older adults following 12 weeks of High-Intensity Interval Training (HIIT) and Moderate-Intensity Continuous Training (MICT).

Parameters	HIIT (*n* = 34)		MICT (*n* = 34)		*p*-Value		
	Pre	Post	Pre	Post	Time Effect	Group Effect	Time*Group Effect
**Anthropometry**							
Weight (kg)	80.2 ± 16.6	80.4 ± 13.7	80.8 ± 18.8	80.3 ± 18.7	0.64	0.94	0.27
BMI (kg/m^2^)	28.9 ± 4.9	29.0 ± 4.9	29.8 ± 6.2	29.6 ± 6.1	0.51	0.59	0.27
**Fat and lean mass (DXA)**							
Total lean mass (kg)	47.4 ± 9.8	48.2 ± 10.2 **	47.2 ± 11.6	46.9 ± 11.4	0.24	0.78	**0.002**
Arms lean mass (kg)	5.6 ± 1.8	5.6 ± 1.8	5.3 ± 1.7	5.1 ± 1.6	0.16	0.40	0.24
Legs lean mass (kg)	16.9 ± 3.7	17.2 ± 3.8 *	16.8 ± 4.1	16.7 ± 4.0	0.28	0.73	**0.03**
Total fat mass (%)	36.9 ± 7.6	36.5 ± 7.5	37.4 ± 7.5	36.9 ± 7.3	**0.03**	0.80	0.83
Arms fat mass (%)	34.6 ± 10.1	33.5 ± 9.7 *	34.2 ± 10.2	35.5 ± 9.8 **	0.85	0.74	**0.0004**
Legs fat mass (%)	34.9 ± 10.3	34.4 ± 10.3	36.4 ± 11.8	34.9 ± 11.3	**0.0001**	0.70	**0.04**
Android fat mass (%)	46.5 ± 7.5	46.1 ± 7.6	46.5 ± 6.4	46.1 ± 6.7	0.21	0.97	0.93
Gynoid fat mass (%)	39.0 ± 9.8	38.7 ± 9.8	40.6 ± 10.9	38.8 ± 10.9 ***	**0.002**	0.74	**0.02**
**Muscle composition (pQCT)**							
Total muscle area (cm^2^)	100.1 ± 29.1	99.1 ± 30.0	104.6 ± 28.8	88.3 ± 26.24 ***	**0.004**	0.74	**0.0004**
Total fat area (cm^2^)	80.2 ± 45.4	78.5 ± 43.3	83.3 ± 42.0	74.1 ± 36.8 ***	**0.002**	0.98	**0.02**
Subcutaneous fat area (cm^2^)	75.1 ± 45.3	74.0 ± 42.7	78.3 ± 41.8	68.9 ± 36.6 ***	**0.002**	0.90	**0.01**
Intramuscular fat area (cm^2^)	5.0 ± 2.1	4.4 ± 2.4	4.8 ± 2.8	3.9 ± 2.0	0.15	0.38	0.90

Data are presented as: mean ± SD. HIIT = high-intensity interval training; MICT = moderate-intensity continuous training; Pre = before the 12-week intervention; Post = after the 12-week intervention; DXA = dual-energy X-ray absorptiometry; pQCT = peripheral quantitative computed tomography; BMI = body mass index; Time effect, Group effect and Time*Group effect were analyzed using two-way repeated measures ANOVA. * *p* < 0.05, ** *p* < 0.01, *** *p* < 0.001 = HIIT effect, and MICT effect (analyzed using post-hoc tests).

**Table 4 healthcare-10-01346-t004:** Blood parameters in obese older adults following 12 weeks of High-Intensity Interval Training (HIIT) and Moderate-Intensity Continuous Training (MICT).

Parameters	HIIT (*n* = 34)		MICT (*n* = 34)		*p*-Value		
	Pre	Post	Pre	Post	Time Effect	Group Effect	Time*Group Effect
**Blood parameters**							
Adiponectin (µg·mL^−1^)	14.5 ± 7.7	13.7 ± 7.6	15.6 ± 8.9	15.4 ± 8.9	0.34	0.53	0.75
Leptin (ng·mL^−1^)	21.6 ± 18.7	24.1 ± 18.9	25.9 ± 21.1	25.1 ± 18.3	0.71	0.52	0.26
Adiponectin/leptin	1.7 ± 2.9	0.9 ± 0.9	1.1 ± 1.2	1.1 ± 1.1	0.20	0.36	0.22
Free fatty acids (mmol·L^−1^)	0.5 ± 0.1	0.5 ± 0.2	0.5 ± 0.1	0.5 ± 0.3	0.40	0.43	0.20
Total cholesterol (mmol·L^−1^)	5.2 ± 1.3	5.1 ± 1.2	4.9 ± 0.9	4.8 ± 1.0	0.33	0.31	0.84
HDL (mmol·L^−1^)	1.4 ± 0.4	1.5 ± 0.4	1.5 ± 0.3	1.5 ± 0.3	0.25	0.94	0.91
LDL (mmol·L^−1^)	3.0 ± 1.1	3.1 ± 0.9	2.8 ± 0.8	2.8 ± 0.8	0.54	0.31	0.70
Triglycerides (mmol)	1.5 ± 0.7	1.3 ± 0.6 **	1.3 ± 0.5	1.3 ± 0.6	**0.03**	0.42	0.09
Ferritin (µg·L^−1^)	136.3 ± 121.8	123.0 ± 101.4 *	91.1 ± 68.1	85.1 ± 66.7	**0.04**	0.08	0.28
IGF1 (µg·mL^−1^)	0.09 ± 0.02	0.09 ± 0.02	0.09 ± 0.03	0.09 ± 0.03	0.74	0.20	0.36
IGFBP3 (µg·mL^−1^)	1.8 ± 0.3	1.9 ± 0.4	1.9 ± 0.4	1.8 ± 0.3	0.80	0.62	**0.002**
IGF1/IGFBP3	0.05 ± 0.01	0.05 ± 0.01	0.05 ± 0.02	0.05 ± 0.01	0.99	0.38	0.44
Glucose (mmol·L^−1^)	5.9 ± 1.3	5.9 ± 1.5	5.6 ± 0.6	5.7 ± 0.8	0.20	0.26	0.70
Insulin (pmol)	45.8 ± 24.5	49.3 ± 30.1	49.2 ± 36.3	46.2 ± 32.3	0.93	0.93	0.20
QUICKI	0.43 ± 0.05	0.42 ± 0.05	043 ± 0.06	0.43 ± 0.05	0.67	0.84	0.73
HOMA-IR (M·U)	2.0 ± 1.2	2.2 ± 1.4	2.1 ± 1.9	2.0 ± 1.7	0.85	0.96	0.38

Data are presented as: mean ± SD. HIIT = High-Intensity Interval Training; MICT = moderate-intensity continuous training; Pre = before the 12-week intervention; Post = after the 12-week intervention; HDL = high-density lipoprotein; LDL = low-density lipoprotein; IGF-1 = insulin-like growth factor-1; IGFBP-3 = insulin-like growth factor binding protein-3; QUICKI = quantitative insulin-sensitivity check index; HOMA = homeostatic model assessment for insulin resistance; M·U = mass unit. Time effect, Group effect and Time*Group effect were analyzed using two-way repeated measures ANOVA. * *p* < 0.05, ** *p* < 0.01 = HIIT effect, and MICT effect (analyzed using post-hoc tests).

**Table 5 healthcare-10-01346-t005:** Skeletal muscle mitochondrial content in obese older adults following 12 weeks of High-Intensity Interval Training (HIIT) and Moderate-Intensity Continuous Training (MICT).

Parameters (A.U)	HIIT (*n* = 11)		MICT (*n* = 14)		*p*-Value		
	Pre	Post	Pre	Post	Time Effect	Group Effect	Time*Group Effect
**Skeletal muscle mitochondrial content**							
OPA1	0.044 ± 0.017	0.047 ± 0.019	0.042 ± 0.020	0.044 ± 0.018	0.63	0.72	0.82
TFAM	0.070 ± 0.025	0.091 ± 0.036 *	0.051 ± 0.030	0.067 ± 0.028 *	**0.003**	0.06	0.64
VDAC	0.123 ± 0.074	0.113 ± 0.068	0.084 ± 0.077	0.096 ± 0.063	0.98	0.31	0.55
MFN1	0.010 ± 0.005	0.012 ± 0.006	0.008 ± 0.006	0.009 ± 0.005	0.15	0.43	0.58
MFN2	0.004 ± 0.003	0.006 ± 0.004 *	0.004 ± 0.004	0.004 ± 0.003	0.39	0.58	0.06
DRP1	0.078 ± 0.031	0.082 ± 0.031	0.084 ± 0.044	0.080 ± 0.036	0.92	0.94	0.58
TOM20	0.113 ± 0.047	0.162 ± 0.058 **	0.101 ± 0.042	0.108 ± 0.023	**0.02**	**0.02**	0.06
PARKIN	0.038 ± 0.024	0.048 ± 0.025 **	0.022 ± 0.013	0.026 ± 0.015	**0.002**	**0.02**	0.12
OXPHOS-CI (NDUFB8)	0.038 ± 0.022	0.048 ± 0.025	0.035 ± 0.025	0.042 ± 0.022	0.30	0.72	0.66
OXPHOS-CII (SDHB)	0.028 ± 0.021	0.033 ± 0.020	0.030 ± 0.025	0.033 ± 0.026	0.50	0.80	0.62
OXPHOS-CIII (UQCRC2)	0.026 ± 0.020	0.028 ± 0.027	0.016 ± 0.024	0.017 ± 0.017	0.56	0.17	0.89
OXPHOS-CIV (MTCO1)	0.004 ± 0.004	0.005 ± 0.006	0.002 ± 0.002	0.003 ± 0.003	0.47	0.15	0.63
OXPHOS-ATPs	0.081 ± 0.064	0.080 ± 0.072	0.063 ± 0.097	0.064 ± 0.074	0.91	0.51	0.85
OXPHOS-TOT	0.178 ± 0.116	0.195 ± 0.131	0.141 ± 0.157	0.160 ± 0.124	0.50	0.46	0.97

Data are presented as: mean ± SD. HIIT = high-intensity interval training; MICT = moderate-intensity continuous training; Pre = before the 12-week intervention; Post = after the 12-week intervention; A.U = Arbitrary Unit. OPA1= optic atrophy-1; TFAM = transcription factor A mitochondrial; VDAC = voltage-dependent anion channel; MFN1 = mitofusin-1; MFN2 = mitofusin-2; DRP1 = dynamin-related protein 1; TOM20 = translocase of outer membrane 20; PARKIN = Parkin RBR E3 ubiquitin protein ligase; OXPHOS-C = oxidative phosphorylation complex; NDUFB8 = NADH: ubiquinone oxidoreductase subunit B8; SDHB = succinate dehydrogenase complex iron sulfur subunit B; UQCRC2 = ubiquinol-cytochrome C reductase core protein 2; MTCO1 = mitochondrially encoded cytochrome C oxidase I; ATPs = adenosine triphosphate synthase; TOT = Total. Time effect, Group effect and Time*Group effect were analyzed using two-way repeated measures ANOVA. * *p* < 0.05, ** *p* < 0.01 = HIIT effect, and MICT effect (analyzed using post-hoc tests).

## Data Availability

The datasets used during the current study are available from the corresponding author on reasonable request.

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
