# Peer review of "Clinical and Biological Adaptations in Obese Older Adults Following 12-Weeks of High-Intensity Interval Training or Moderate-Intensity Continuous Training"

_healthcare, 2022, doi:10.3390/healthcare10071346_

Round 1
Reviewer 1 Report
1. Why didn't you set a negative control group other than HIIT and MICT group.
2. How did authors get this sample size? Is the sample size appropriate for the study? Did authors calculate it?
Author Response
We thank the reviewer for taking the time to read the manuscript and we are thankful for her/his constructive comments.
Please see the attachment.

Reviewer 2 Report
The authors aimed to elucidate the differences between the impact of HIIT and MICT on functional capacities, muscle function, body composition, blood biomarker, gene expression in the adipose tissue and markers of muscle mitochondrial content in obese older adults. Although they showed a broad range of data, there are a few comments on the manuscript.
First, is there any possibility that older people may respond to both types of exercise differently compared to younger people in metabolic adaptation? Older people may need longer period of time to express the changes in functional and/or biological parameters in response to the same degree of exercise compared to younger people, i.e. 3 months of intervention that could generally lead to certain alterations was enough and sufficient to induce some significant changes in older people?
Second, the participants in both groups completed over 95% of exercise sessions in the study. Though the structured, supervised exercise sessions were definitely effective and useful, the evaluation of patient-reported-outcome should be necessary to promise widespread dissemination of efficacious exercise and its sustainability in the population. The authors should comment on that in the context of improving QOL.
Third, the authors claimed HIIT could be time-efficient and recommended to obese older adults in Abstract and Conclusion. But older people may have more spare time and it should be highlighted that they should get more active with reduced sedentary lifestyle and that both HIIT and MICT can be useful options to be recommended to obese older people whichever they like.
Author Response

(The authors gave the same response as above.)

Reviewer 3 Report
The article represents interesting research regarding adaptations in obese adults following HIIT or MICT training. It is very relevant and it should be of great interest to the readers. Here are my comments and suggestions:
Abstract
Abstract is nicely written with all the relevant information.
Introduction
Introduction chapter contains all the relevant information, theoretical background and clearly stated aim of the study.
Methods
If the study combines two randomized controlled trials, were they approved by the same ethics board, under the same number? Furthermore, were the trials registered? If yes, state the body of registration and registration number. RCTs should be registered.
One of the inclusion criteria was being inactive which the authors defined as having less than 2 hours of structured physical activity per week. This would mean that individuals with 119 minutes of structured physical activity per week are considered inactive which might not be the true. Regarding the exclusion criteria, you mention metal implant like pacemaker. Metal implant is also TKA or THA. Please, be more specific regarding the type of metal implants which are considered exclusion criteria.
How did you match participants into groups? How did estimate sample size?
Please, provide information regarding the manufacturer of the exercise equipment (company, country). Please provide the references for the physical performance tests (validity and reliability). When you mention reference Buckinx et al. (2018) [28 ] it seems redundant to put year of publication in the bracket besides number of reference in square brackets. What type of hand dynamometer did you use? Provide model besides manufacturer. In what position did you measure hand grip strength?
Did you check normality of the data? If yes, what test did you use?
Results
Results are clearly written.
Discussion
I suggest not to mention Tables in the Discussion chapter. I would advise to expand the discussion with more comparisons of the trial with previous studies.
Conclusion
Conclusion is adequate and does not need changes.
Author Response

(The authors gave the same response as above.)
